# DEGS: Deformable Event-based 3D Gaussian Splatting from RGB and Event Stream

## Abstract

Reconstructing Dynamic 3D Gaussians Splatting (3DGS) from low-framerate RGB videos is challenging. This is because large inter-frame motions will increase the uncertainty of the solution space. For example, one pixel in the first frame might have more choices to reach the corresponding pixel in the second frame. Event cameras can provide super-fast visual change acquisition asynchronously while not containing color information. Intuitively, the event stream can provide deterministic constraints for the inter-frame large motion by the event trajectories. Hence, combining low-temporal resolution images with high-framerate event streams can address this challenge. However, the data format of the two modalities is very different, and currently, no methods directly optimize dynamic 3DGS from events and RGB images. This paper introduces a novel framework that jointly optimizes dynamic 3DGS from the two modalities. The key idea is to adopt event motion priors to guide the optimization of the deformation fields. First, we extract the motion priors encoded in event streams by using the proposed LoCM unsupervised fine-tuning framework to adapt an event flow estimator to a certain unseen scene. Then, we present the geometry-aware data association method to build the event-Gaussian motion correspondence, which is the primary foundation of the pipeline, accompanied by two useful strategies, namely motion decomposition and inter-frame pseudo-label. Extensive experiments show that our method outperforms existing image and event-based approaches across synthetic and real scenes and prove that our method can effectively optimize dynamic 3DGS with the help of event data.

## 1 Introduction

High-quality dynamic scene reconstruction from a monocular video is significant for various applications, such as AR/VR, animation modeling, computer graphics, and 3D content creation. Previous approaches Cao & Johnson (2023); Gao et al. (2021); Guo et al. (2022); Liu et al. (2023); Pumarola et al. (2021); Wu et al. (2023); Yang et al. (2023a) extend the conventional static reconstruction methods, such as Neural Radiance Field (NeRF) and 3D Gaussian Splatting (3DGS), into temporal dimensions. These methods require spatial and temporal dense input to model the dynamical process. Therefore, they cannot faithfully reconstruct dynamic scenes when the number of video frames is insufficient and sparse, which might be caused by the fast-deforming objects or low frame rate RGB cameras.

The event camera, also known as the dynamic vision sensor (DVS), works very differently from traditional cameras. Instead of capturing full frames at regular intervals, the event camera only detects relative changes in brightness for each pixel independently and asynchronously. This allows it to capture rich information about objects' movements and deformations, leading to several advantages including high temporal resolution, high dynamic range, and reduced data storage. Therefore, the event camera can handle the challenges mentioned above. Pure event streams do not contain absolute radiance information. Fortunately, conventional event cameras generally have an APS sensor, which captures RGB images at a lower frame rate. In this case, the event stream and RGB images complement each other; the event stream provides rich information on relative changes between frames, while the sparse RGB images provide absolute color information. This work proposes to combine the two modalities for dynamic scene reconstructions.

Previous studies Hwang et al. (2023); Rudnev et al. (2023); Wang et al. (2024) attempt to reconstruct 3D static scenes from pure event streams by incorporating the linearized event generation model into the NeRF pipeline. However, they cannot either model dynamic scenes or optimize NeRF from the two modalities. Hence, they can only synthesize gray-scale static novel views. Very recently, DE-NeRF Ma et al. (2023) first adopted event and RGB data to optimize the dynamic NeRF. This method asynchronously estimates per-event color by using RGB and the event generation model and then uses these estimated colors along with sparse RGB images to jointly optimize the dynamic NeRF. Nevertheless, it fails to explore the rich inter-frame motion information in the event stream, resulting in a suboptimal reconstruction. On the other hand, even though NeRF delivers good results in neural 3D reconstruction, the original NeRF suffers from large training and rendering costs. Recent 3DGS Kerbl et al. (2023) significantly boosts the rendering speed to a real-time level by replacing the cumbersome volume rendering in NeRF with efficient differentiable splatting. Moreover, 3DGS can also produce higher-fidelity rendering results. Very recently, Ma et al. (2023) and Wu et al. (2023) equipped 3DGS with a deformation field to model dynamic scenarios. However, these approaches still rely on the high framerate of image sequences. When the frame rate of the video is low, the motion between images is too large, which can lead to the inability to optimize the Gaussian motion trajectory modeled by the deformation field. To the best of our knowledge, currently, there is thus no method to use event cameras for optimizing dynamic 3D Gaussian.

In this work, we propose a novel framework that can efficiently optimize the Dynamic 3D Gaussian Splatting from the two modalities, namely event stream and sparse RGB images. The primary obstacle is the large motion between sparse images, which prevents the deformation field from being easily optimized. The key idea of this work is that the event trajectories on the 2D plane can be used to optimize the deformation field in the dynamic 3DGS because the edge motion of objects triggers events. First, we present a LoRA-based unsupervised framework to finetune an event flow predictor, which can restrict our correction not to overturning the original priors and adapt it to unobserved scenarios. Second, we propose a geometry-aware method to build event-Gaussian data associations, which is the primary core for our optimization task. Next, we adopt the motion decomposition and inter-frame pseudo-label strategies to perform better.

Our main contributions can be summarized as follows:

- We propose the LoCM unsupervised finetuning framework by leveraging low-rank adaptation and contrast maximization to adapt a pretrained event flow estimator to an unseen scenario while maintaining its original priors.

- To take full advantage of the motion cues in the event stream, we propose the geometry-aware data association method that can build motion correspondence between 2D events and 3D Gaussians to utilize event trajectories to optimize the deformation field.

- To mitigate the dynamic ambiguity, we propose to use a motion decomposition scheme and inter-frame pseudo labels to assist optimization.

- We establish a novel synthetic event-based dataset and we build a pipeline to convert commonly used image-based real-world datasets into the event-based version by using a real event camera DVXplorerIni (2024) to facilitate the community. Experiments prove the effectiveness of optimizing dynamic scenes by the two modalities.

## 2 RELATED WORK

### 2.1 DYNAMIC NOVEL VIEW SYNTHESIS

Synthesizing novel views of a dynamic scenario from captured 2D sequences remains a challenge. Since NeRFMildenhall et al. (2020) has achieved great success in novel view synthesis, many efforts have been made to generalize NeRF to capture dynamic scenes. Pumarola et al. (2021); Du et al. (2021); Li et al. (2021); Liu et al. (2023); Park et al. (2021a) combine NeRF with additional time dimension or time-conditioned latent codes to reconstruct time-varying scenarios. Fang et al. (2022); Guo et al. (2023); Yi et al. (2023); Fridovich-Keil et al. (2023); Gan et al. (2023); Li et al. (2022); Shao et al. (2023); Wang et al. (2023) explicitly incorporate voxel grids to model temporal information. Additionally, Cao & Johnson (2023); Song et al. (2023); Abou-Chakra et al. (2023) attempts to construct explicit structures to learn a 6D hyperplane function without directly modeling.

Recently, a novel point-based representation, i.e. 3DGS, has been presented which formulates points as 3D Gaussians with learnable parameters. Although the vanilla 3DGS regards the scene as static, a few works have attempted to extend 3DGS to dynamic scenes due to its real-time rendering and high reconstruction quality. D-3DGS Luiten et al. (2023) is the first attempt to adopt 3DGS to dynamic scenes. Inspired by dynamic NeRFs, Wu et al. (2023); Yang et al. (2023b;a); Yan et al. (2024) introduces the deformed-based 3DGS that preserves a set of canonical Gaussians and learns the deformation field at each timestep. These works introduce the topological invariance into the training pipeline, thereby enhancing their suitability for the reconstruction of dynamic scenarios from monocular inputs. Besides, some other works Xie et al. (2023); Guo et al. (2024); Feng et al. (2024); Zhong et al. (2024) choose to explicitly formulate the continuous motion for deformation using the assumption that the dynamics of the scene are the consequences of the movement. In this work, our method proposes to establish a highly efficient pipeline for optimizing dynamic 3DGS from event streams and sparse RGB images.

## 2.2 EVENT-BASED NEURAL RECONSTRUCTION

Traditional RGB cameras suffer from motion blurs when the moving speed of cameras is fast and cannot be used in extreme lighting conditions because of their low dynamic ranges. However, most existing 3D reconstruction methods fail to provide event-based solutions. To address this issue, several works Klenk et al. (2023); Hwang et al. (2023) have been proposed to incorporate the linearized event generation and NeRF pipeline. However, these works fail to reconstruct clear edges and textures and suffer from soft fogs caused by continuous NeRF networks. PAEv3D Wang et al. (2024) introduces motion priors into the NeRF pipeline, enhancing the quality of edge and texture reconstruction. EvGGS Wang et al. introduces the first optimization framework to combine multiple event-based vision tasks with 3DGS. These previous works all regard the scenario as static. Nevertheless, Ma et al. (2023) is the first to introduce event streams to model dynamic radiance fields and achieve satisfactory results. As we mentioned above, the sparsity and discontinuity of event data could lead to blurs and soft fogs because NeRF encodes scenes as a continuous network. To reconstruct the dynamic scene more faithfully, we incorporate event-based data and 3DGS and utilize the event flow extracted from the event stream for optimization. We experimentally prove that our reconstruction quality outperforms existing image-based and image-based methods.

## 3 METHODOLOGY

### 3.1 DYNAMIC 3D GAUSSIAN SPLATTING

Standard 3DGS represents the scene using 3D Gaussian points, each of which is characterized by several trainable parameters including position ($\mu \in \mathcal{R}^3$), quaternion ($\mathbf{q} \in \mathcal{R}^4$), scale factor ($\mathbf{s} \in \mathcal{R}^3$), spherical harmonics coefficients ($\mathbf{h} \in \mathcal{R}^{3(k+1)^2}$) and opacity ($o \in [0, 1]$). The quaternion defines a $3 \times 3$ rotation matrix ($\mathbf{R}$). The 3D covariance matrix can be obtained by $\Sigma = RSS^T R^T$. Given a certain camera pose, one can render the novel view by projecting these 3D Gaussians into a 2D plane by blending depth-ordered Gaussians overlapping that pixel.

$$C = \sum_{i \in \mathcal{N}} c_i \alpha_i \prod_{j=1}^{i-1} (1 - \alpha_j) \tag{1}$$

where $c_i$ refers to the color of the Gaussian $i$. $\alpha_i$ denotes the soft occupation of Gaussian $i$ at the certain 2D location, which can be obtained by :

$$\alpha_i(x) = o_i exp(-\frac{1}{2}(x - \mu_i)^T \Sigma_i^T (x - \mu_i)) \tag{2}$$

In Eq. 2, $\Sigma$ and $\mu$ specifically refer to their 2D-projected version while they keep their original 3D meanings in other equations. The reconstruction loss between renderings and groundtruth images is used to optimize all Gaussian parameters.

The conventional 3DGS only focuses on static scene reconstruction. Due to its flexible and explicit features, it is easy to be extended to reconstruct 4D dynamic scenarios. The most intuitive way is to separately train multiple 3D-GSs in each timestep and then interpolate between these sets (dynamic 3DGS tracking). However, it falls short of continuous monocular captures within a temporal

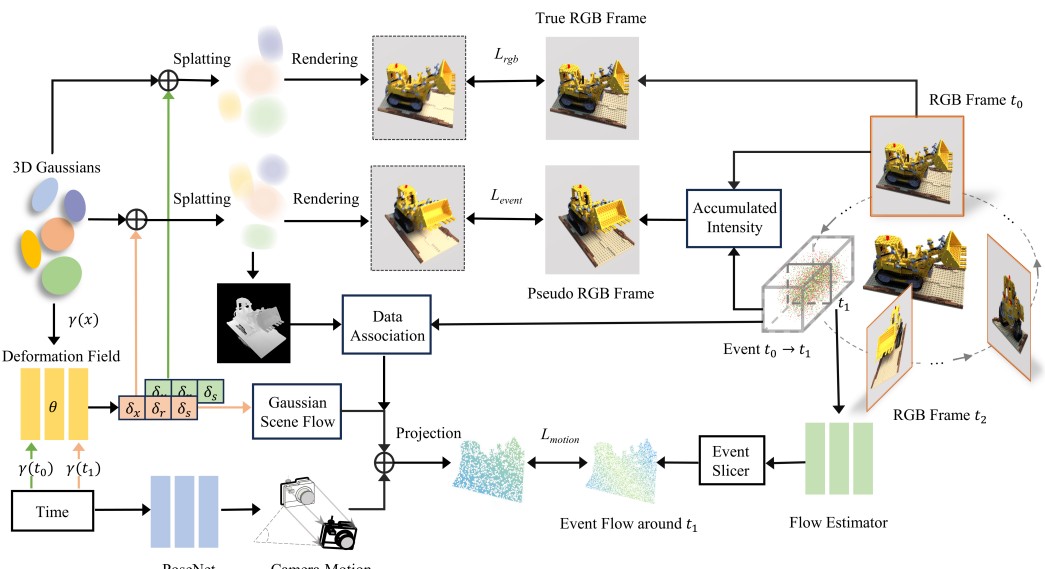

Figure 1: The overview of our proposed framework.

sequence and leads to excessive memory consumption. A more prevailing way in the community is to jointly learn a deformation field along with the canonical 3D Gaussians. The deformation field transforms each canonical Gaussian point's position, rotation, and scale parameters into the corresponding values at the target timestamp, shown in Eq. 3.

$$(\delta\mathbf{x}, \delta\mathbf{s}, \delta\mathbf{q}) = \mathcal{F}_\theta(\gamma(\mathbf{x}), \gamma(\mathbf{t})) \tag{3}$$

where $\gamma$ denotes the positional encoding. The representative works are Deform-GS Yang et al. (2023b) and 4D-GS Wu et al. (2023), which can effectively learn 4D scene representation from monocular videos. The Deform-GS adopts a neural network to implement $\mathcal{F}$ while 4D-GS uses a triplane representation for that. In our work, we follow the Deform-GS to utilize a neural network to model the deformation field.

**Problem Statement**. The frame-based dynamic 3DGS can be optimized from high frame-rate videos. However, it cannot be optimized from event camera data, i.e. the high temporal resolution event stream accompanied by sparse image sequences. Given a video captured by an event camera, including the event stream $[\mathbf{E_k}]_1^k$ and sparse RGB $[I_n]_1^N$, our goal is to reconstruct the high-quality dynamic scene from the two modalities.

**Method Overview**. The primary reason why optimizing dynamic 3DGS from videos with low temporal resolution is difficult can be summarized as follows. Large deformation between two frames causes the solution ambiguity. For example, the number of paths to reach the same point in the second frame from the corresponding point in the first frame will be uncertain. In this case, the deformation field cannot be easily optimized. However, the inter-frame information provided by event cameras, i.e. the event trajectories, can be used to guide the training of the deformation field. Therefore, the key issues become 1. how to determine the event trajectories and 2. how to build motion correspondence between 2D event data and 3D Gaussian points. Since the motion of Gaussians in the same area can be considered similar, training for their scale and rotation parameters is not a major issue Wang et al..

## 3.2 EXTRACTING PRIORS FROM EVENTS

We introduce how to extract event trajectories in this section. We start with the work principle of the event camera. An event camera carries independent pixel sensors that operate continuously and generate "events" $e_k = (\mathbf{x}_k, t_k, p_k)$ whenever the logarithmic brightness at the pixel increases or decreases by a predefined threshold. Each event $e_k$ includes the pixel-time coordinates ($\mathbf{x}_k = (x_k, y_k), t_k$) and its polarity $p_k = \{+1, -1\}$. The most intuitive thought is that the events are triggered by the motion of the edge information in the scenes. The event trajectories are defined

as the associations between events across different timestamps, which can be represented by the spatial-temporal event flow. Zhu et al. (2018; 2019); Gehrig et al. (2021); Hagenaars et al. (2021) can predict event flow by receiving raw event data as input. These models have been trained on various large datasets with per-event groundtruth. However, their performance would degrade when they are applied to unobserved scenarios, especially those with different motion patterns. A common way to use these pretrained event flow estimators is to finetune them on unseen scenes, but finetuning struggles with two issues, 1. the lack of groundtruth of event flow and 2. overfitting on small dataset overturns the original motion priors in these models, which are learned from large datasets

To address the two challenges, we present to combine low-rank adaptation Hu et al. (2021a) with Contrast Maximization framework Shiba et al. (2022b); Gallego et al. (2018) to finetune the estimator in an unsupervised manner. The scene-specific finetune can be expressed as:

$$W_\theta := W_0 + \Delta W_\theta \tag{4}$$

where we implement the $\Delta W_\theta$ with LoRA at $\Delta W_\theta = BA$. We adopt a random Gaussian initialization for $A$ and zero for $B$, so $\Delta W_\theta = BA$ is zero at the initialization of training. Zero initialization can restrict the correctness not deviating so much from the original value. Moreover, using LoRA can help the correction not to overturn the motion priors.

The Contrast Maximization (CM) framework Gallego et al. (2018); Shiba et al. (2022a;b); Stoffregen et al. (2019) is an effective proxy for extracting motion information from raw event data. The idea behind this optimization framework is that events triggered by the same portion of a moving edge can be wrapped by motion models to produce a sharp event accumulated image. We simply introduce the mathematical foundations of CM as follows. Assume a set of events $\mathbf{E} = e_k \frac{N}{k=1}$ are given, the goal of CM is to obtain the point trajectories on the image plane, which are described by the per-event optical flow. The image of warped events (IWE) can be obtained by aggregating events along candidate point trajectories to a reference timestamp.

$$\mathbf{x}_k^{'} \doteq \mathbf{W}(\mathbf{x}_k, t_k; \theta) = \mathbf{x}_k - (t_k - t_{ref})\mathbf{v}_\theta(\mathbf{x}_k, t_k) \tag{5}$$

Here $\mathbf{W}$ is the warp function based on the event optical flow field $\mathbf{v}_\theta$. If the input flow is correct, this reverses the motion in the events, and results in sharper event images. Thus the objective function is often defined to maximize the contrast of the IWE, given by the variance:

$$Var(IWE(\mathbf{x}; \theta)) \doteq \frac{1}{|\Omega|} \int_\Omega (IWE(\mathbf{x}; \theta) - \mu_I)^2 d\mathbf{x}$$
$$\mu_I = \frac{1}{|\Omega|} \int_\Omega (IWE(\mathbf{x}; \theta)) \tag{6}$$

The objective function measures the goodness of fit between the events and the candidate motion curves (warp). However, if we directly adopt the above IWE as the unsupervised loss, this might encourage all events to accumulate to several certain pixels. Inspired by Shiba et al. (2022a), we adjust the objective by measuring event alignment using the magnitude of the IWE gradient to overcome the issue. Finally, the squared gradient magnitude of the IWE should be set to:

$$Var(\theta; t_{ref}) \doteq \frac{1}{|\Omega|} \int_\Omega ||\nabla IWE(\mathbf{x}; t_{ref}, \theta)||^2 d\mathbf{x} \tag{7}$$

Instead of directly optimizing the IWE, which is sensitive to the arrangement (i.e., permutation) of the IWE pixel values, the optimization target can be regarded as the variance of the IWE. Moreover, To mitigate overfitting, we divide the image plane into a tile of non-overlapping patches and up/down-sampled to multi-scale branches Shiba et al. (2022a). One can iteratively optimize the objective function to obtain the proper parameters $\theta$ of the event flow field. However, in this work, we adopt a flow estimator to predict the event flow instead of directly optimizing parameterized warp functions. The reason is that the pre-trained estimator has already learned prior knowledge on various motion patterns, such as simultaneous camera ego-motion and multi-object motions within the scene, enabling it easier to model unseen complex motion patterns. On the contrary, motion fields composed of simple parameters can hardly distinguish different motion patterns in the scene. From Eq. 7, we can derive the unsupervised learning loss as

$$L_{flow} = 1/Var(f_\theta(\mathbf{E}(t_1, t_{ref}))) + \lambda TV(\theta) \tag{8}$$

where $f_\theta$ is the flow prediction network, $\mathbf{E}(t_1, t_r e f)$ is the events between $t_1$ and $t_r e f$. $\lambda$ denotes the weight to balance the CM unsupervised loss and the $TV$ regularization Rudin et al. (1992). The $f_\theta$ predicts the event flow from $t_1$ to $t_r e f$ of all events in $\mathbf{E}(t_1, t_r e f)$. Maximizing the contrast is equivalent to minimizing its reciprocal. Detailed architecture of the network is in **Appendix**.

By adopting such an unsupervised paradigm and LoRA, we can efficiently finetune the event flow estimator on the specific scene that we aim to reconstruct. The predicted event flow is, in the following, used to guide the optimization of the deformation field.

### 3.3 Event-Gaussian Data Association

In this section, we introduce how to build the data association between 2D events and 3D Gaussians, which is fundamental for optimizing 3DGS with the help of event streams. As discussed in Sec. 3.1, the core challenge is the uncertainty caused by the large motions between frames. In other words, the $\delta x$ in Eq. 3 is hard to optimize, which can be considered as the scene flow of each 3D Gaussian. To align the motion of events with the deformation of 3D Gaussians, we first establish the data associations between events and their corresponding 3D Gaussians. The differentiable renderer of 3DGS can smoothly produce the depth map for a given camera viewpoint at a certain timestamp because the $\alpha$-blending in rendering naturally deals with transparency and occlusion relationship. For a specific timestamp $t$, we produce its depth map at first. Since dynamic 3DGS experiences a warm-up stage (3.1), the depth map can also be initialized well in a coarse manner. Then the 3D location to trigger each event can be found by unprojecting the depth value of each event to 3D space. The nearest 3D Gaussian to the unprojected 3D location is treated to be the most contributory to this event. In addition, we implement the binding correspondence in a soft manner for better robustness. In detail, each 3D Gaussian corresponds with its top k-nearest unprojected events (k=3 empirically) and generates a correspondence weight via the inverse distance weight function for them. At present, the one-to-k Gaussian-to-Events data associations are established by using the geometry information obtained from 3DGS rasterization.

### 3.4 Decomposed Motion Supervision

However, the motion contributing to events contains not only the Gaussian deformation but also the camera ego-motion. Therefore, we decompose the motion signal into the Gaussian scene flow and the camera ego-motion. Unlike other NeRF-based approaches to interpolate camera trajectory or use turnable poses, we follow Ma et al. (2023) to use a PoseNet to generate a continuous pose function that maps time to the camera pose vector representation $(R, t)$. The PoseNet is proved to be very effective and a plug-play module in DE-NeRF Ma et al. (2023). It is fed with the interpolated camera poses as inputs and outputs a corrected term to make it perfect. It is efficient to be trained because even though we directly adopt the interpolated camera poses, we can still obtain relatively not-bad results.

In this context, given two nearby specific poses, we can easily derive their instantaneous translation and rotation velocities ($\mathbf{v}_c = [v_x, v_y, v_z]$ and $\mathbf{w}_c = [w_x, w_y, w_z]$) by assuming the camera is moving rigidly in a small time interval. The pixel-level image velocity caused by the camera ego-motion can be derived from the camera's rigid motion and its corresponding 3D location:

$$\mathbf{F}_{ego} = \frac{1}{Z} \begin{pmatrix} -1 & 0 & x \\ 0 & -1 & y \end{pmatrix} \mathbf{v}_c + \begin{pmatrix} xy & -1-x^2 & y \\ 1+y^2 & -xy & -x \end{pmatrix} \mathbf{w}_c \tag{9}$$

The detailed step-by-step derivation of the above equation can be found in Mitrokhin et al. (2019); Zhu et al. (2019; 2018). The $\mathbf{v}_c$ and $\mathbf{w}_c$ are projected to the image plane given the inverse depth ($\frac{1}{Z}$ in Eq. 9) rendered from the 3DGS rasterization. Likewise, the Gaussian scene flow between two selected timestamps can be derived by:

$$\begin{aligned} \mathbf{F}_{gs}^i = &Proj(\mathbf{x_i} + \mathcal{F}_\theta(\gamma(x_i), \gamma(t_1))[0]) \\ &- Proj(\mathbf{x_i} + \mathcal{F}_\theta(\gamma(x_i), \gamma(t_0))[0]) \end{aligned} \tag{10}$$

Here the $[0]$ refers to selecting the first output term of the deformation field $\mathcal{F}_\theta$. The $Proj$ function denotes the projection of a 3D location to the 2D image plane by camera transformation matrix. $i$ refers the $i$-th Gaussian. Until now, we have established the decomposed motion correspondence from 3D to 2D. The combination of $\mathbf{F}_{gs}$ and $\mathbf{F}_{ego}$ can be optimized through the finetuned event flow estimator presented in Sec.3.2, which we will introduce in detail in the following section.

3.5 TRAINING PARADIGM

This section introduces the detailed training pipeline to optimize dynamic 3DGS by events and RGB modalities simultaneously. We first initialize the Gaussian point set in a warm-up phase with 3500 iterations by merely using the sparse RGBs. This is typically supervised by $L_{rgb} = ||I_c - R(c)||_1^1$ where $c$ is the camera pose. In the second joint training phase, the $L_{rgb}$ remains unchanged. More importantly, we integrate events into the training pipeline by our previously presented approach. At each iteration step, two consecutive images and the event between them are read out by the dataloader. Here we refer to the timestamp of the two images as $t_0$ and $t_1$. Next, we randomly select a timestamp between $t_0$ and $t_1$, which is referred to as $t_e$, for the subsequent event-part optimization.

We leverage the events to optimize the timestamps between two adjacent RGBs. In detail, at each optimization step, in addition to the RGB, we randomly select another timestamp between the image and its next image, i.e. $t_e$. Then we use intermediate events between the image to estimate a pseudo image at the $t_e$ and use it to optimize the 3DGS. The detailed loss that uses the event intensity priors for supervising is as follows:

$$L_{event} = \begin{cases} ||R(t_e) - I(t_0)e^{\Sigma_{e \in \Delta t} p_i C}||_1^1, & t_e \leq \frac{t_0 + t_2}{2} \\ ||R(t_e) - I(t_2)/e^{\Sigma_{e \in \Delta t} p_i C}||_1^1, & t_e > \frac{t_0 + t_2}{2} \end{cases} \tag{11}$$

$t_0$ and $t_2$ refer to the timestamps of the left and right RGB images. To mitigate the occlusion effect, we use the nearest RGB to estimate the color at $t_1$.

Moreover, we adopt the flow estimator ($f_\theta(\mathbf{E}; t_{ref})$ in Eq. 8) to estimate the event trajectories, i.e. event optical flow, from $t_1$ to $t_0$. The $t_{ref}$ in this phase should be constantly set to the timestamp of the nearest RGB in Eq. 11. Here for the convenience of illustration, we assume $t_e < \frac{t_0 + t_1}{2}$ so as $t_{ref} = t_0$. As we discussed in the previous section, the event flow is triggered by two decomposed motions. After applying the flow estimator, the optical flow of all events between $t_0$ and $t_1$ can be resolved. We then bind these events near $t_0$ to their corresponding $k$ 3D Gaussians based on the approach introduced in Sec. 3.3. We use a small time interval to filter out candidate events ($t_1 - \delta t \leq t_e \leq t_1$). Only considering the events whose timestamps are near $t_0$ is necessary because the same entity might trigger independent events at different timestamps. Each Gaussian corresponds to at most $k = 3$ events and at least 0 events. Next, we jointly train the ego-motion PoseNet and Gaussian deformation by :

$$L_{motion} = \sum_{i=1}^{N} (\sum_{k \in \mathcal{B}(i)} f_\theta(\mathbf{E}(t_0, t_1)) - (\mathbf{F}_{ego} + \mathbf{F}_{gs}^i)) \tag{12}$$

where $i$ denotes the $i$-th Gaussians with at least one associated event. $\mathcal{B}(i)$ illustrates the bond operation to establish event-Gaussian correspondence. Other symbols remain the same in the previous context. We set $k = 3$ experimentally. After the warm-up phase, the total loss is:

$$L_{total} = L_{rgb} + \gamma_1 L_{event} + \gamma_2(iter)L_{motion} \tag{13}$$

$\gamma_1$ and $\gamma_2$ are hyperparameters to balance the magnitude of these items. We set $\gamma_1 = 1$ and $\gamma_2 = 1 - e^{-\frac{1}{4000}iter}$ that is a function of training iterations. At the early training stage, the deformation field cannot predict accurate Gaussian scene flow. We use an annealing function to weigh this term.

## 4 EXPERIMENTS

In this section, we thoroughly validate the effectiveness of the proposed approach both on the synthetic and real-world datasets and ablate the components constituents contained in this approach.

**Synthetic Dataset.** On account of the absence of relevant synthetic event-based monocular 4D reconstruction dataset. We establish a novel one with three scenarios. They carry varying degrees and types of large-scale motions. We adopted Blender to produce video camera poses and simulated event streams using V2E Hu et al. (2021b).

**Realistic Dataset.** We evaluate our methods and other counterparts on three real-world datasets. The first is the High-Speed Events and RGB dataset (HS-ERGB) Tulyakov et al. (2021) which includes challenging and moving fastly dynamic scenes captured by a realistic event camera. We select three

| Methods | Centaur | | | Lego | | | Man | | |
|---|---|---|---|---|---|---|---|---|---|
| | PSNR↑ | SSIM↑ | LPIPS↓ | PSNR↑ | SSIM↑ | LPIPS↓ | PSNR↑ | SSIM↑ | LPIPS↓ |
| HyperNeRF | 33.09 | 0.9814 | 0.0256 | 28.94 | 0.9545 | 0.0487 | 33.25 | 0.9758 | 0.0281 |
| Deform-GS | 33.74 | 0.9829 | 0.0187 | 30.88 | 0.9646 | 0.0350 | 34.98 | 0.9805 | 0.0221 |
| 4DGS | 33.53 | 0.9819 | 0.0194 | 31.34 | 0.9662 | 0.0322 | 37.60 | 0.9817 | 0.0184 |
| DE-NeRF | 37.35 | 0.9895 | 0.0149 | 33.30 | 0.9780 | 0.0261 | 35.87 | 0.9815 | 0.0194 |
| Ours | 42.15 | 0.9971 | 0.0076 | 36.00 | 0.9916 | 0.0120 | 39.37 | 0.9856 | 0.0149 |

Table 1: Quantitative comparisons of baselines and ours on synthetic scenes.

| Methods | Candle | | | Umbrella | | | Fountain | | |
|---|---|---|---|---|---|---|---|---|---|
| | PSNR↑ | SSIM↑ | LPIPS↓ | PSNR↑ | SSIM↑ | LPIPS↓ | PSNR↑ | SSIM↑ | LPIPS↓ |
| HyperNeRF | 35.72 | 0.9489 | 0.2512 | 27.83 | 0.8379 | 0.2687 | 22.85 | 0.8036 | 0.5013 |
| Deform-GS | 37.12 | 0.9579 | 0.2266 | 32.77 | 0.8611 | 0.1557 | 26.38 | 0.8516 | 0.4688 |
| 4DGS | 36.95 | 0.9569 | 0.2299 | 32.44 | 0.8598 | 0.1617 | 27.12 | 0.8717 | 0.4547 |
| DE-NeRF | 37.05 | 0.9546 | 0.2405 | 32.38 | 0.8609 | 0.1591 | 26.84 | 0.8523 | 0.4345 |
| Ours | 37.77 | 0.9450 | 0.1132 | 34.07 | 0.8513 | 0.1210 | 30.54 | 0.9125 | 0.0277 |
| Methods | Banana | | | Chicken | | | Chocolate | | |
| | PSNR↑ | SSIM↑ | LPIPS↓ | PSNR↑ | SSIM↑ | LPIPS↓ | PSNR↑ | SSIM↑ | LPIPS↓ |
| HyperNeRF | 22.70 | 0.5167 | 0.3441 | 24.66 | 0.7589 | 0.2990 | 24.66 | 0.8452 | 0.1697 |
| Deform-GS | 22.65 | 0.5067 | 0.3294 | 25.07 | 0.7621 | 0.1959 | 28.66 | 0.9041 | 0.1218 |
| 4DGS | 22.49 | 0.4958 | 0.3353 | 25.06 | 0.7613 | 0.1936 | 25.25 | 0.8470 | 0.1466 |
| DE-NeRF | 24.60 | 0.6206 | 0.4636 | 27.61 | 0.8323 | 0.2344 | 27.36 | 0.8916 | 0.1689 |
| Ours | 31.68 | 0.9586 | 0.0950 | 30.17 | 0.9371 | 0.1526 | 30.29 | 0.9460 | 0.1240 |

Table 2: Quantitative comparisons of baselines and ours on real-world datasets.

challenging scenes in the dataset, i.e. **Umbrella**, a rapid rotating scene, **Candle**, an environment with rapid jitter, and **Fountain** a super-fast liquid scenario. The dataset provides complete and realistic event streams, RGB sequences as well as camera parameters. The other two real-world datasets that we use in this work, the HyperNeRF dataset Park et al. (2021b) and the NeRF-DS dataset Yan et al. (2023), are commonly used in purely image-based NeRF or 3DGS research. They only provide RGB videos but unfortunately without corresponding real event streams. Using simulators for event generation makes it difficult to replicate the various noises found in events captured by real event cameras. Therefore, we use a realistic event camera, DVXplorer Ini (2024) to convert the two datasets into their event-based versions to obtain realistic images and event data pairs for the convenience of comparison with conventional image-based methods. The details of the data collection pipeline and collection system setup are illustrated in **Appendix**.

## 4.1 IMPLEMENTATION DETAILS

We adopt the EvFlowNet Zhu et al. (2018) as the $f_\theta$ in Eq. 8. We first load its original network parameters then we use Eq. 4 and Eq. 8 to finetune it to fit a single scene event stream. We use Adam to optimize the EvFlowNet with an exponential decay learning rate from 0.0005 to 0.0001.

## 4.2 RESULTS AND COMPARISONS

In our experiments, we compare our method with four baselines, HyperNeRF Park et al. (2021b), 4DGS Wu et al. (2023), Deformable GS (Deform-GS) Yang et al. (2023b), and DE-NeRF Ma et al. (2023). The first three methods are currently prevailing image-based dynamic reconstruction approaches while the last one is the SOTA event-based dynamic NeRF method. We perform both quantitative and qualitative evaluations for a comprehensive and convincing comparison.

**Quantitative Comparison.** Firstly, we quantitatively evaluate these methods on scene reconstruction quality metrics including PSNR, SSIM, and LPIPS. We employ the VGG network for LPIPS evaluation. As shown in Table 1 and Table 2, we compute the mean values for all metrics across scenes from both synthetic and realistic datasets. We use two symbols to indicate the top two performing methods in nine scenarios, with Coral , Orange and Yellow representing the best, second best, and third best, respectively, and second best respectively. The results demonstrate that our proposed approach exhibits superior performance across all scenarios, proving the effectiveness of our framework.

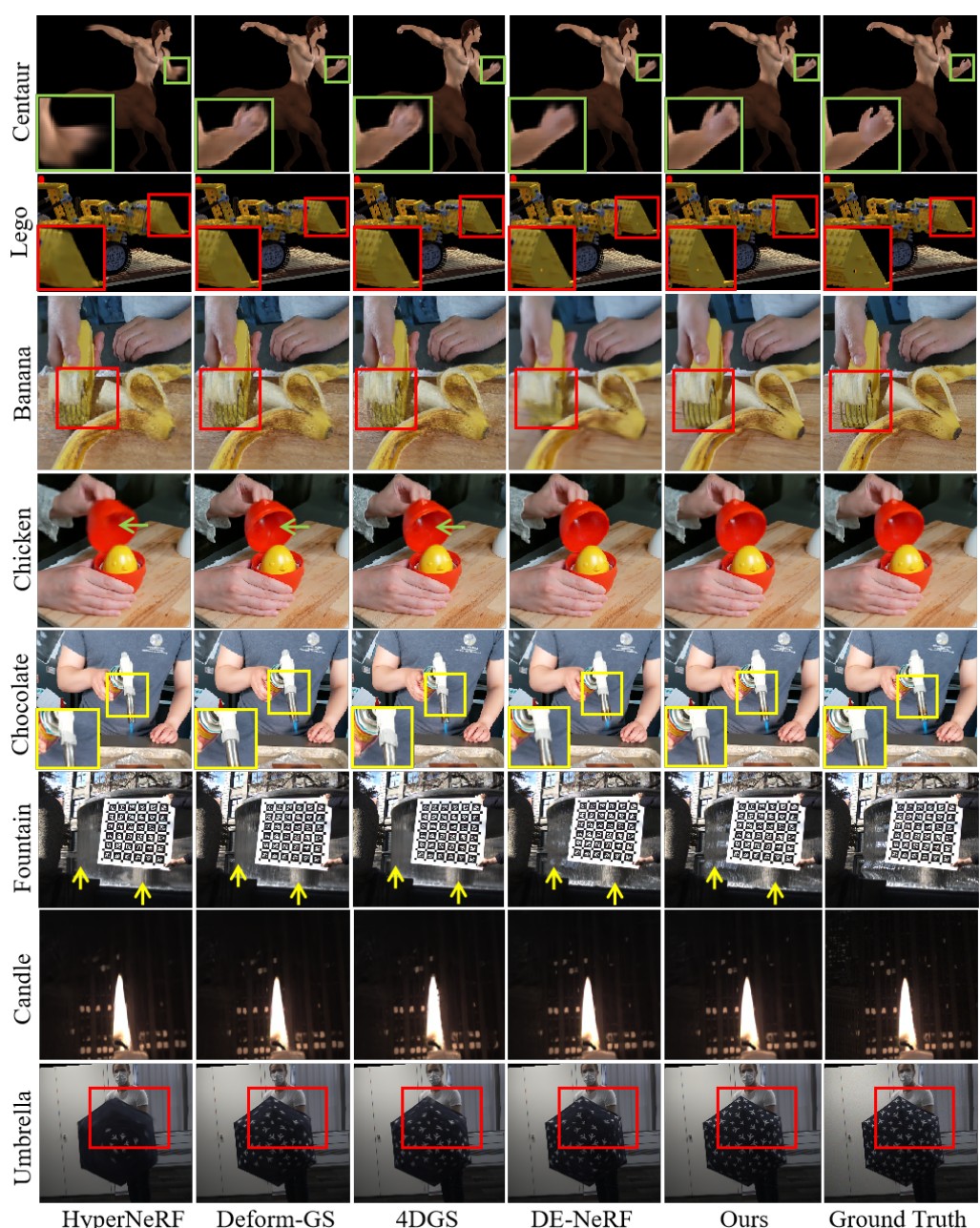

Figure 2: **Qualitative Comparison on synthetic and real-world datasets.** Regions with notably different reconstruction qualities are highlighted with colored boxes and arrows.

**Qualitative Comparison.** We also provide qualitative results and comparisons for a better visual assessment. We visualize the results in Fig. 2, and it can be observed that our method recovers more detailed information when synthesizing images from novel viewpoints. Our method reconstructs more delicate object contours and textures, especially in the regions annotated with boxes. Our approach surpasses existing methods by a great margin, demonstrating the remarkable capability of restoring the contents and details of the given scene over time. The effectiveness of our method benefits from the rich information derived from both RGB image modality and event modality.

## 4.3 ABLATION STUDY

We report the contributions of each part of the training configurations and components of the proposed approach in this section. As shown in Table 3, "only $L_{rgb}$" denotes that our method degrades

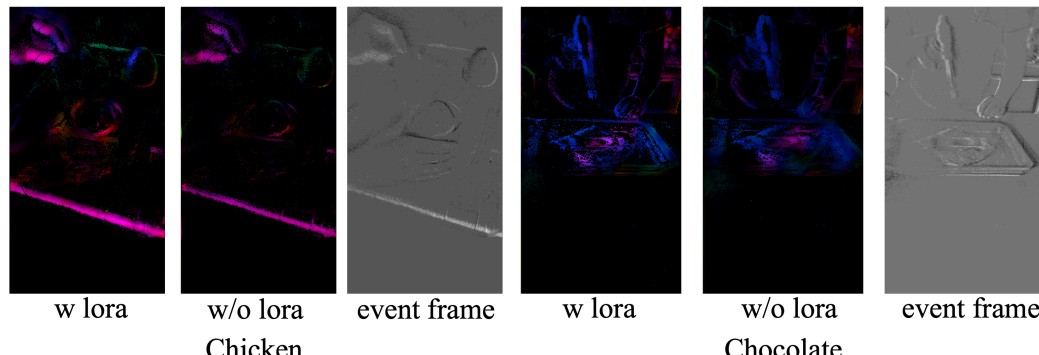

| | | | | | |
|---|---|---|---|---|---|
| w lora | w/o lora | event frame | w lora | w/o lora | event frame |
| | Chicken | | | Chocolate | |

Figure 3: Qualitative examples of event flow prediction with or without LoCM fine-tuning.

| | Lego | | | Chocolate | | |
|---|---|---|---|---|---|---|
| | PSNR↑ | SSIM↑ | LPIPS↓ | PSNR↑ | SSIM↑ | LPIPS↓ |
| only $L_{rgb}$ | 30.80 | 0.964 | 0.035 | 28.66 | 0.904 | 0.122 |
| w/o $L_{event}$ | 31.62 | 0.966 | 0.031 | 28.92 | 0.917 | 0.133 |
| w/o $L_{motion}$ | 31.24 | 0.968 | 0.031 | 25.65 | 0.861 | 0.183 |
| w/o Flow Ft | 32.47 | 0.973 | 0.025 | 27.31 | 0.892 | 0.174 |
| Full Ft | 34.61 | 0.981 | 0.026 | 29.11 | 0.915 | 0.168 |
| w/o PoseNet | 34.15 | 0.977 | 0.022 | 28.62 | 0.906 | 0.157 |
| unaltered | **34.89** | **0.982** | **0.021** | **30.29** | **0.946** | 0.124 |

Table 3: Ablations studies of different components in the proposed framework.

to the conventional deformable GS and only uses sparse RGBs to train the model. "w/o $L_{event}$" means $\gamma_1 = 0$ in Eq. 13, while "w/o $L_{motion}$ refers that $\gamma_2$ is constantly set to 0 in Eq. 13. "w/o Flow Ft" represents that we directly adopt the original weights that are open-source with its code of EvFlowNet to estimate event flow without using Eq. 8 to finetune it. "Full Ft" refers to that we normally fine-tune all parameters in the FlowNet without using the LoRA technique. "w/o PoseNet" means we use pose interpolation between frames instead of the PoseNet to get the event pose, even though this has already been proved to be effective by Ma et al. (2023). We validate all the components on the three scenes by comparing their respective PSNR, SSIM, and LPIPS. The results illustrate that all the above modules contribute differently to learning dynamic Gaussian fields from event data. Among these, the motion prior has a greater impact on the results, while the color prior and pose have less influence.

Moreover, we indicate two visual examples to show the superiority of the LoCM fine-tuning strategy in Fig. 3. It is observed that the predicted flows fine-tuned by LoCM ("w lora" in the figure) are sharper, and can better distinguish the object boundaries. In addition, they exhibit strong contrast between objects. Notably, quantitative results are not reported because our dataset, as we stated previously, is recorded by a realistic DVXplore, thereby no groundtruth of event-level flows. More comparison and analysis can be found in **Appendix**.

## 5 CONCLUSION

In this work, for the first time, we introduce a novel framework that can effectively reconstruct the dynamic 3DGS from two modalities, i.e. event stream and sparse RGB images. The success of this approach can be attributed to the following design. We adopt the inter-frame motion priors encoded in event data to optimize the dynamic 3DGS. We extract the motion priors by a flow estimator that is finetuned by the proposed LoCM unsupervised finetuning framework to produce event trajectories. We decompose the entire motion into the 3DGS deformation and the camera ego-motion and use two independent to predict them. The two networks can be optimized by the extracted event trajectories. In addition, we adopt the knowledge of intensity changes between frames contained in event data to supervise the renderings where there is no existing image. To validate the methods, we create our synthetic dataset and convert two datasets conventionally used in image-based dynamic scene reconstruction into the event-based version by a realistic DVXplorer event camera. Experiments show that our method outperforms both existing image-based and event-based approaches on all datasets by a large margin.

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

## A    IMPACT STATEMENTS

This work introduces a novel framework that can optimize Dynamic 3D Gaussian Splatting from two modalities, i.e. sparse RGB images and event streams. Event camera has the advantages of high dynamic range and high temporal resolution. Thus, in fast-moving scenes or extreme lighting conditions, event cameras have great potential. This work provides insights to solve the above challenges by using event cameras in reconstructing dynamic scenes.

## B    EVENT REPRESENTATION

In this section, we give a detailed introduction to the representation of asynchronous raw event streams and how we preprocess them as input for the optical flow estimator. Events ($e_k = (\mathbf{u}_k, t_k, p_k)$) at pixel $\mathbf{u}_k = (u, v)$ and timestamp $t_k$ are triggered and output asynchronously, and the illumination change of the pixel can be represented using the polarity ($p \in \{+1, -1\}$). Thus, an event triggered at timestamp $t_k$ can be written as :

$$\Delta L_k(\mathbf{u}) = \sum_{e_i \in \Delta t_k} p_i C \tag{14}$$

Where L is the logarithmic frame ($L(t) = log(I(t))$) and C denotes the event trigger threshold value. Therefore, given threshold value $C$ and time interval $\Delta t$, we could accumulate events triggered in $\Delta t$, and thus obtain the log illumination changes. Due to the sparsity of the event streams, we need to convert the asynchronous event streams at the given time interval $\Delta t$ to a synchronous representation. Thus. the event streams are encoded as a spatial-temporal voxel grid. The duration $\Delta t$ is divided into $B$ temporal bins following

$$E(u, v, t_n) = \sum_i p_i \max(0, 1 - |t_n - t_i^*|) \tag{15}$$

where $t_i^*$ is the normalized timestamp determined by the number of bins by $\frac{B-1}{\Delta T}$. We set the temporal bins $B = 5$.

## C    DATA COLLECTION SYSTEM

This section introduces our device setup for data collection. The data collection pipeline aims to produce corresponding event data for an RGB video. In this work, except for the HS-ERGB Tulyakov et al. (2021), which contains complete and realistic event streams of high-speed scenes, the other two datasets we utilize, the Hypernerf dataset Park et al. (2021b) and the NeRF-DS dataset Yan et al. (2023), are image-based datasets, which only contain RGB video sequences. An intuitive method is to utilize an event camera simulator to convert the video into an event stream, as we have done in synthetic data. However, the synthetic event stream is not realistic enough. The main reason is that the event captured by a realistic event camera contains lots of irregular noises due to fluctuations of electronic components such as hot noises which do not exist in synthetic data. Even though one can add handcrafted noises to the simulated event stream, such as Gaussian Noises, they are still far from reality. To address this, we build the data collection system (see Fig. 4). In the system, we use a well-calibrated realistic event camera (DVXplorer) and a high fresh rate screen with a 300 fresh rate to convert RGB videos with 120 FPS to real event streams.

First of all, we align the camera with the high refresh rate screen, ensuring that the camera aligns with the top left corner of the RGB video by using the Checkerboard alignment. Then, we replay the original videos with high temporal resolution on the high-frame screen. At the same time, we adopt the DVXplore to capture the screen to produce event data. The entire system is located in a no-light workspace, ensuring that the screen is the only light source, which can largely reduce or even eliminate the impact of screen reflection. Events captured by this pipeline could contain realistic features such as irregular noises, we illustrate this point in Fig. 5. In this figure, the left side is the input video, and the right upper is the synthetic event data, which are clean and regular. The right lower panel is the event stream generated by our system, which is dense, irregular, and noisy. There are obvious differences between synthetic and real events. Furthermore, we record the original videos again with a low temporal rate (1/5 of the original temporal resolution) because

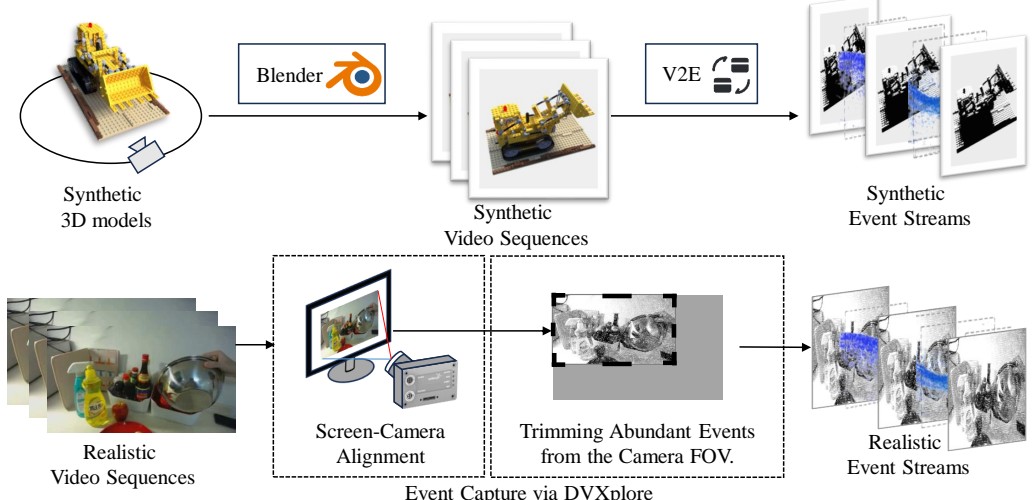

Figure 4: Data collection system. We leverage the realistic DVXplorer event camera and a high frame rate screen to convert the HyperNeRF and NeRF-DS datasets into their event-based version.

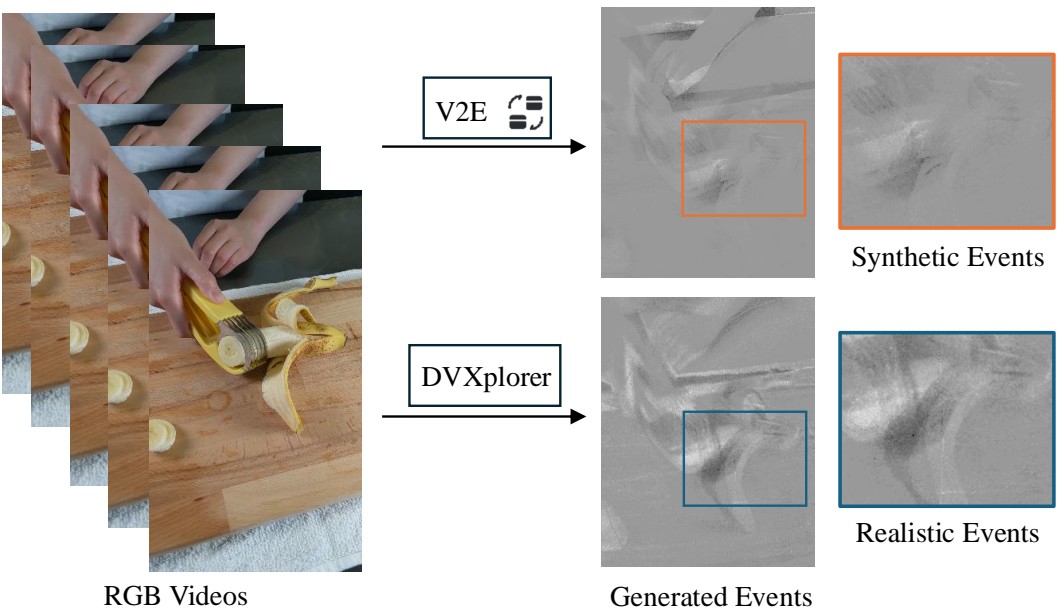

Figure 5: Visualization results of synthetic and realistic events of Banana scene

the Active Perception Sensors (APS, which is armed on some premium-version event cameras and used to capture colored images) of the event camera usually have a low frame rate, and we want to simulate the phenomenon. The proposed data collection system is cheap and efficient and can be used to convert any RGB video into the corresponding realistic events. The advantages of the setup include but are not limited to 1. One does not have to go outside to find various scenes to create a high-quality dataset. In contrast, they can fully utilize rich RGB video resources on the Internet 2. Overcome the low fidelity of the event camera simulator 3. One does not require a premium-version event camera (with an APS sensor), instead, our pipeline only needs a fundamental version event camera, such as DVXplorer, which is cheaper.

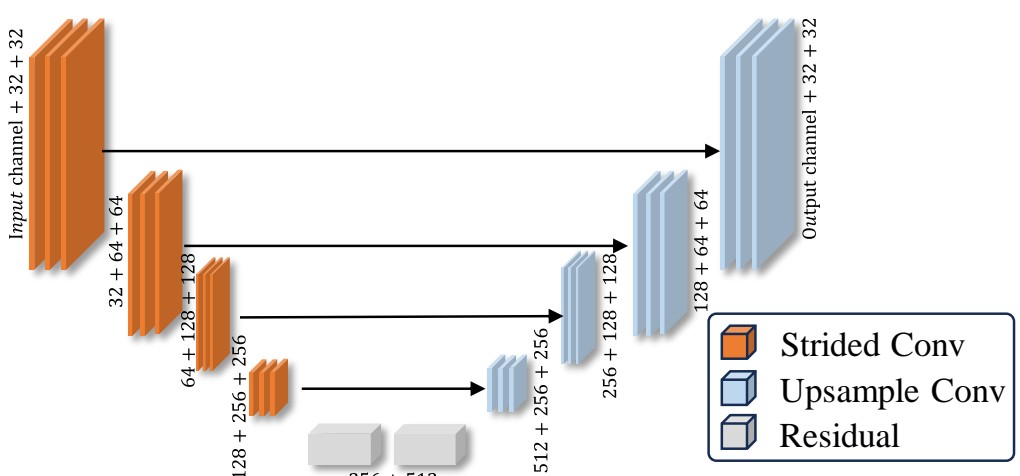

Figure 6: **The architecture of the event-based optical flow estimator**

## D    DETAILED NETWORK ARCHITECTURES

In this section, we introduce the detailed architectures and parameter selections. The networks include the deformation network to transform canonical 3D Gaussians, the PoseNet to map time to generate continuous poses, and the optical flow estimator to generate the event-based optical flow. The deformation network is learned using an MLP network $\mathcal{F}_\theta$. This deformation network transforms the canonical position, rotation, and scale to the corresponding value given the target timestamp. The MLP receives the input and passes it to an 8-layer fully connected layers that employ the ReLU function as activation and feature 256-dimensional hidden layers and outputs a 256-dimensional feature vector. The vector is then passed through three additional prediction heads to predict the position, rotation, and scale of the 3D Gaussians. Similar to NeRF, there is a skip connection between the input feature vector and the fourth layer. Unlike DENeRF Ma et al. (2023), which uses an 8-layer MLP, a more lightweight MLP is utilized in our proposed method. Our PoseNet architecture only contains the sinusoidal encoder and a 2-layer MLP to map time $t$ to translation and rotation speed $(v, w)$. The network receives normalized time $t \in \mathbf{R}$ as input and output $(v, w)$ following Rodrigues's formula.

The architecture of the event-based optical flow estimator is very similar to the U-Net networks. The framework receives the event spatial-temporal voxel grid as input and consists of the stridden convolution encoder, two residual block layers, and the upsample convolution decoder with skip connections to the corresponding encoder layer. We visualize the network structure in Fig.6. The monocular event stream passes the downsample convolution encoders. The tanh function is applied as the activation function, and the features are passed to the residual blocks and then upsampled four times using the nearest neighbor resampling for the flow estimation.

## E    COMPARISONS ON NERF-DS DATASET

We also quantitatively evaluate our method and four baselines on the NeRF-DS dataset and conduct the evaluation experiment with metrics such as PSNR, SSIM, and LPIPS.

The results are presented in Table.4 and indicate that our method achieves outstanding performance in all metrics and most of the scenes within the dataset. Cells are marked as **bold** and underline, representing the best, and second best respectively.

## F    DEPTH VISUALIZATION

As illustrated in Fig.7, we visualize the depth map of test scenes. Our proposed method yields substantially more accurate depth maps than other baselines, highlighting its superior geometric

| Methods | Cup | | | As | | | Press | | |
|---|---|---|---|---|---|---|---|---|---|
| | PSNR↑ | SSIM↑ | LPIPS↓ | PSNR↑ | SSIM↑ | LPIPS↓ | PSNR↑ | SSIM↑ | LPIPS↓ |
| HyperNeRF | 23.55 | 0.8529 | 0.1986 | 26.69 | 0.9175 | 0.1573 | 25.38 | 0.8549 | 0.1978 |
| Deform-GS | 23.87 | 0.8591 | 0.1807 | 26.75 | 0.9189 | 0.1590 | 25.72 | 0.8735 | 0.2022 |
| 4DGS | 23.81 | 0.8601 | 0.1794 | 26.59 | 0.9165 | 0.1580 | 25.75 | 0.8655 | 0.1917 |
| DE-NeRF | 24.25 | 0.8708 | 0.1791 | 26.24 | 0.9065 | 0.1962 | 24.51 | 0.8448 | 0.2382 |
| Ours | 24.10 | 0.8740 | 0.1728 | 27.18 | 0.9224 | 0.1581 | 25.82 | 0.8691 | 0.1891 |

Table 4: Quantitative comparisons on NeRF-DS dataset. We also compare our method against previous methods on three real-world scenes from the NeRF-DS dataset.

reconstruction capabilities. This underscores the method's efficacy across both synthetic and real-world datasets.

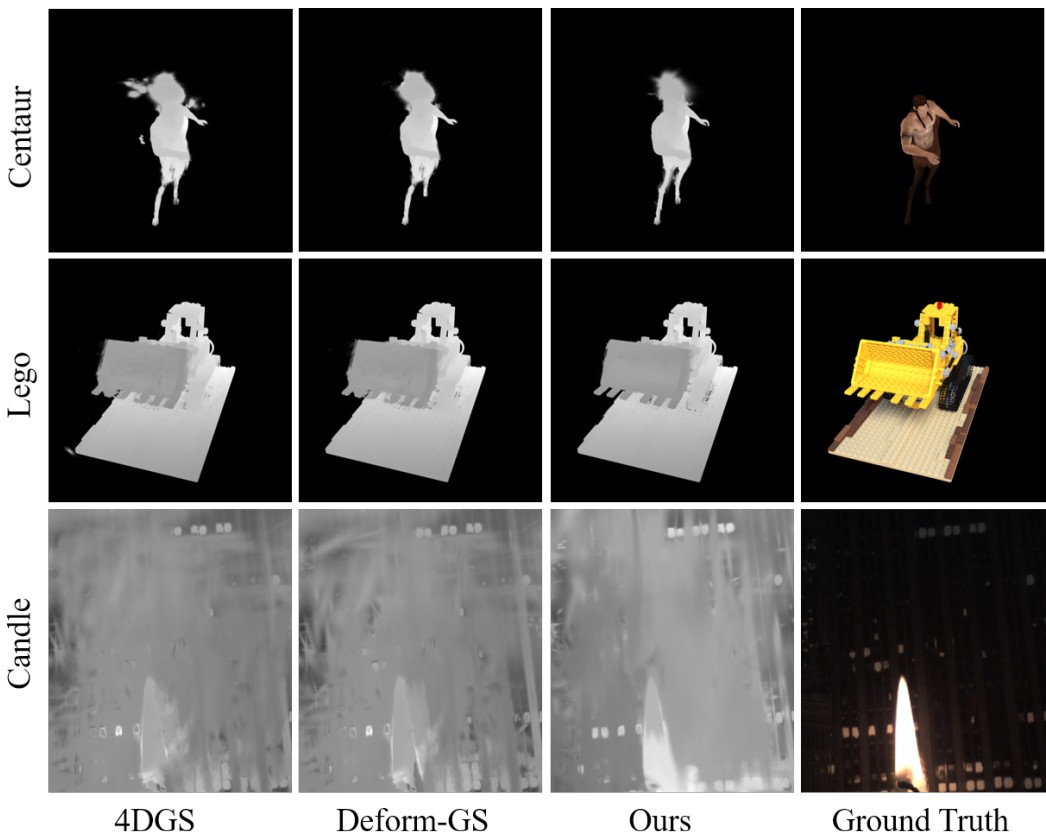

Figure 7: Depth visualization. We compare our method against 4DGS Wu et al. (2023) and Deform-GS Yang et al. (2023b) on both synthetic and real-world scenes. These scenes are Centaur, Lego, and Candle from the top down.

## G ADDITIONAL QUALITATIVE COMPARISON RESULTS

As shown in Fig.8, We also provide additional qualitative results on novel scenes and previously evaluated scenes from new viewpoints. Certain regions in the images are magnified to compare the recovered details and demonstrate the differences in reconstruction quality between our method and other baselines. In addition, we provide **video demonstrations** of our method and other 3DGS-based single modality approaches in the **Supplementary Material**. These materials support that the proposed method can effectively optimize dynamic 3DGS from both RGB images and event streams, especially when the RGB images are sparse.

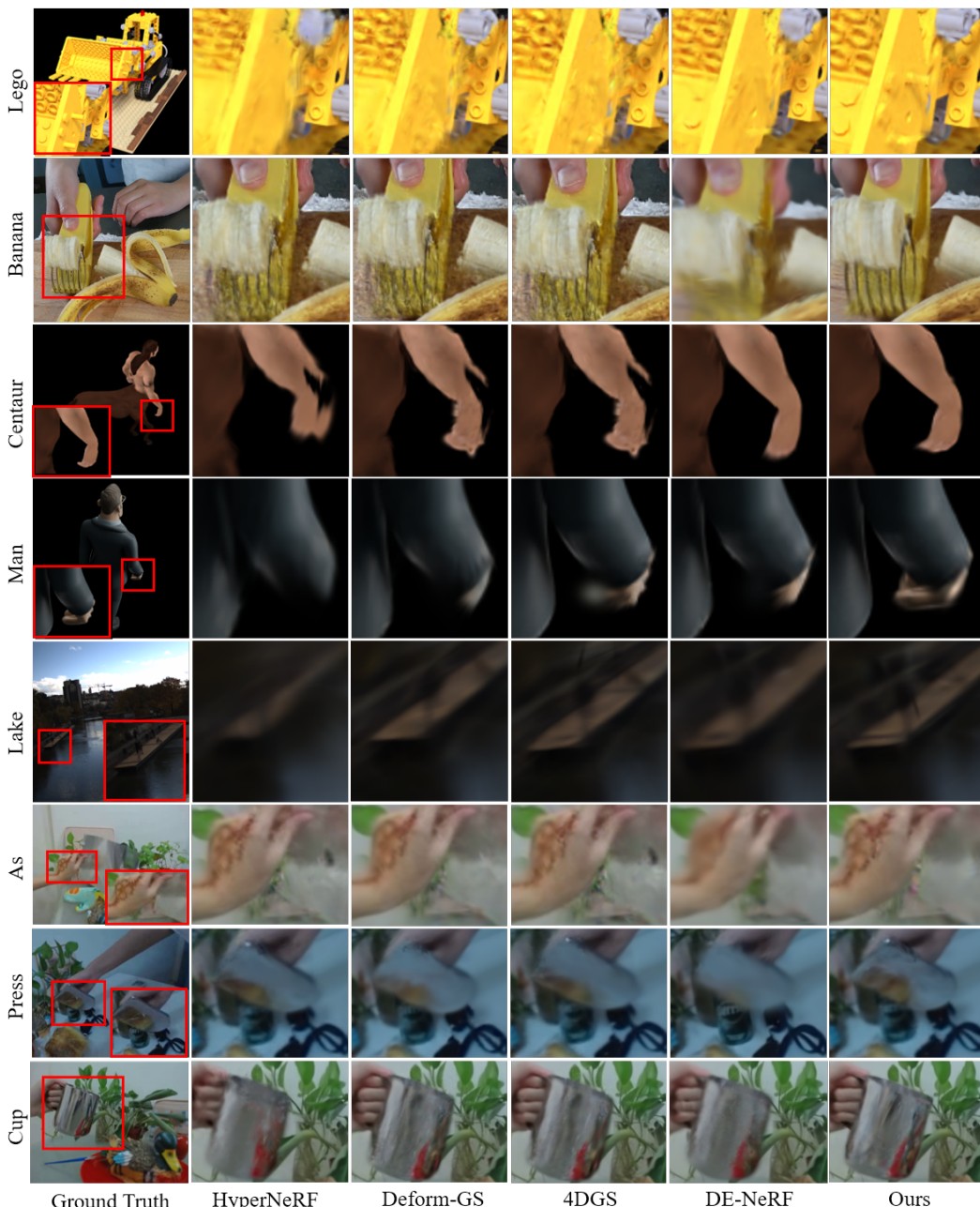

Figure 8: **Comparison visualization based on enlarged images.** The Cup, Press and As scenarios are from the NeRF-DS dataset, and Lake is another scene from the HSERGB dataset. Lego and Centaur are our novel synthetic datasets. Banana is from the HyperNeRF dataset.

## H  LIMITATIONS

Although a large improvement has been achieved, this work has some limitations. This approach relies on the precomputing to extract the event flow as the motion prior which is used to guide the training of the deformation field. The pertaining of the event motion estimator block will take some time and have some biases. In the future, we plan to incorporate the motion extraction block into the whole 3DGS training pipeline. In this case, we can simultaneously train the two blocks and make them benefit from each other mutually.

