# OpenReview forum: "DEGS: Deformable Event-based 3D Gaussian Splatting from RGB and Event Stream"
_ICLR.cc/2025/Conference — ICLR 2025 Conference Withdrawn Submission_

### Official Review · Reviewer_xonS · 2024-10-28

**Soundness:** 4
**Presentation:** 3
**Contribution:** 3
**Rating:** 6
**Confidence:** 4

**Summary:**

To address the challenge of low 3DGS reconstruction quality in dynamically deformable scenes and to obtain accurate scene representation, DEGS performs multi-modal fusion of rich event signals from time cameras and sparse RGB signals. The focus is on leveraging event signals as a guiding signal to describe scene dynamic deformations, optimizing the learning of the deformation field. DEGS creatively proposes the LoCM module, a training module for event flow prediction that combines LoRA and contrast maximization, providing an event flow prior for deformation field optimization. The rendered depth map is used as a medium to establish a connection between 2D events and 3DGS units through inverse projection. Like many previous methods, DEGS decomposes overall motion into camera ego-motion and deformation motion of the GS units when handling movement. Ultimately, by combining event and RGB signals, it achieves SOTA results in dynamic scene GS reconstruction.

**Strengths:**

Originality:
Firstly, DEGS is an innovative attempt to enhance the quality of dynamic scene representation in 3DGS-based methods using event signals. Event cameras have significant advantages in dynamic representation, and the field of 3DGS representation needs an approach like DEGS. Furthermore, DEGS creatively combines methods such as LoRA and CM, allowing them to function effectively in the right context, providing valuable prior guidance for dynamic deformation.

Quality:
To validate DEGS's performance, multiple datasets were used, including those real-world datasets without event signals. For datasets lacking event signals, DEGS proposed a solution using DVXplore recordings to compensate for this gap. The results in the DEGS paper also demonstrate its effectiveness across multiple dynamic scene datasets. Compared with various baselines, DEGS shows superior performance in certain details.

Clarity:
The overall writing of DEGS is relatively clear, making it easy for readers familiar with the 3DGS field to understand its intended message. DEGS also provides concise introductions to some foundational methods referenced, enhancing the ease of reading and comprehension.

Significance:
The representation of dynamic scenes has always been an important task, as the real-world environment we live in is a constantly changing dynamic scene. AI is also evolving, from understanding information in static images to comprehending video signals, where capturing details in dynamic scenes is crucial. In the field of 3DGS, modeling dynamic scenes is essential, and effectively integrating the capabilities of multiple sensors is a valuable direction for exploration.

**Weaknesses:**

(1)The article frequently mentions low-framerate RGB as a premise. When RGB is sparse, low-quality 3DGS is a common phenomenon, which becomes more pronounced in dynamic scenes. However, DEGS does not clearly indicate whether the sparsity of RGB is a result of comparison with event signals or if it is simply a result comparison conducted in an environment with limited RGB data. A more detailed explanation of the results is expected.
(2)Although DEGS achieves good scene representation results, much of the framework's design and usage relies on combining existing methods. The core of its approach is not an original design that breaks the framework of previous methods. As DEGS mentions in the conclusion, the current design has limitations and is not a tightly integrated fusion.

**Questions:**

(1)Inter-frame consistency is an important metric in dynamic scene representation. When I reviewed the video visualization results, I clearly observed the inconsistency exhibited by the highlighted red areas in the "Chicken" dataset. Of course, this issue is also evident in the comparative baseline. I hope to gain more explanations regarding consistency and to understand the potential and limitations of DEGS in this aspect.
(2)In terms of experiments, regarding the performance on the real dataset (Table 2), it is clear that the results in the second row show significant improvement compared to those in the first row. Please provide more experimental explanations to clarify this phenomenon. Additionally, the discussion of DEGS's impact on the experimental results is insufficient. I would like to know if this is related to the performance during the DVXplore capture process?
(3)Are there any plans for open sourcing in the future?

---

### Official Review · Reviewer_AGYZ · 2024-11-03

**Soundness:** 2
**Presentation:** 2
**Contribution:** 2
**Rating:** 3
**Confidence:** 5

**Summary:**

This paper proposes a dynamic scene reconstruction method using event streams and RGB frames. The current event-based method, which is based on NeRF, suffers from large training and rendering costs. This paper introduces a 3DGS-based framework that optimizes deformable 3DGS using event stream and RGB images. To explore the rich inter-frame motion information in the event stream, it presents a LoRA-based unsupervised framework to finetune an event-based optical flow predictor. It also proposes a geometry-aware method to build event-Gaussian data associations. The proposed approach outperforms SOTA on three synthetic scenarios and six realistic scenarios.

**Strengths:**

1. It proposes the LoCM unsupervised finetuning framework to explore the rich inter-frame motion information in the event stream.
2. It introduces the geometry-aware data association method to build event-Gaussian motion correspondence, and use inter-frame pseudo labels to assist optimization.

**Weaknesses:**

1. The contribution of the proposed method is limited. The main goal of the proposed method is to accelerate the training speed of the previous event-based methods and reduce rendering costs. The most significant contribution to achieving this goal is to change the baseline (Deformable NeRF [R1]) of DE-NeRF to Deformable 3DGS, which appears to lack novelty.
2. The proposed unsupervised finetuning of the event flow predictor is unreliable. In section 3.2, the authors use equation 8 to finetune EV-flownet. However, the text lacks further analysis or related work to support that maximizing the variance of predicted optical flow (which differs from CM loss [R2]) could facilitate achieving more accurate flow prediction. A comprehensive study with convincing validation is necessary.
3. The details of the experiments lack clarity. The authors should explain how to create the train-test sets in the nine scenarios to ensure that the test set is captured in a novel view. It is worth noting that the three realistic scenarios from the HS-ERGB dataset were recorded using only one static camera, which implies that they cannot be utilized to evaluate the performance of novel view synthesis.
4. The writing quality of this paper is poor. Several crucial parts of the exposition are unclear. For example, why is it necessary to build event-Gaussian association in section 3.3 (since you can directly use optical flow to optimize the warp field without the association)? The utilization of symbols in section 3.2 (e.g., t_{r}ef), section 3.4 (e.g., the Proj function whose timestamp corresponding to the camera transformation matrix is not specific), and section 3.5 (e.g., the timestamp t1 and t2) is also confusing.


[R1] Park K, Sinha U, Barron J T, et al. Nerfies: Deformable neural radiance fields[C]//Proceedings of the IEEE/CVF International Conference on Computer Vision. 2021: 5865-5874.

[R2] Shiba S, Aoki Y, Gallego G. Secrets of event-based optical flow[C]//European Conference on Computer Vision. Cham: Springer Nature Switzerland, 2022: 628-645.

**Questions:**

1. Why is the result of ‘unaltered’ in Table 3 different from the result of ‘ours’ in Table 2?
2. How do you get the result of DE-NeRF since its code is not released?

---

### Official Review · Reviewer_2KBD · 2024-11-03

**Soundness:** 2
**Presentation:** 2
**Contribution:** 2
**Rating:** 5
**Confidence:** 4

**Summary:**

The paper introduces a novel framework for reconstructing dynamic 3D scenes using low-frame-rate RGB videos combined with high-frame-rate event streams. This approach addresses challenges in dynamic scene reconstruction by utilizing the unique strengths of event cameras.

Key Contributions
1) Unsupervised Finetuning Framework: Introduces a LoCM framework for adapting event flow estimators to new scenes while preserving original motion priors.
2) Geometry-Aware Data Association: Develops a method to correlate 2D events with 3D Gaussian motions, leveraging event trajectories to optimize deformation fields.
3) Motion Decomposition and Pseudo-Labeling: Utilizes a decomposition scheme to assist in dynamic scene optimization.

**Strengths:**

* The integration of high-frame-rate event data with RGB offers a robust solution for dynamic scene reconstruction.
* The framework effectively handles large motions and low frame rates, which are common challenges in this field.

**Weaknesses:**

* The evaluations on real datasets could be more thorough. Why only three scenes are chosen from HS-ERGB dataset? How about the performance on other sequences?

* The results on real datasets are not always the best, as seen in Table 4, even the proposed method exploit additional event measurements. The proposed method performs only slightly better, or sometimes worse, than other methods. It would be helpful for the authors to analyze the reasons. Providing more results on real data could offer further insights.

* The presentation in Figure 1 could be further improved, in the part on how event measurements interact with the other parts.

**Questions:**

none

---

### Official Review · Reviewer_oZv4 · 2024-11-04

**Soundness:** 2
**Presentation:** 2
**Contribution:** 1
**Rating:** 1
**Confidence:** 5

**Summary:**

This work builds upon the deformation-field-based dynamic 3DGS methods [1, 2], incorporating high-temporal-resolution event data into the optimization pipeline to provide temporally dense motion supervisions, thereby enhancing reconstruction quality. The authors integrate an event-based optical flow estimator as part of the pipeline to predict 2D optical flow from event data. Using an unsupervised LoRA-based framework, they fine-tune the event flow estimator to adapt to unseen scenes while preserving the original priors. Subsequently, by applying projection and motion decomposition, they associate 3D Gaussians with 2D optical flow estimates, allowing the event data to supervise the deformation field in 3D space.

[1] Wu, G., Yi, T., Fang, J., Xie, L., Zhang, X., Wei, W., ... & Wang, X. (2024). 4d gaussian splatting for real-time dynamic scene rendering. In Proceedings of the IEEE/CVF Conference on Computer Vision and Pattern Recognition (pp. 20310-20320).

[2] Yang, Z., Gao, X., Zhou, W., Jiao, S., Zhang, Y., & Jin, X. (2024). Deformable 3d gaussians for high-fidelity monocular dynamic scene reconstruction. In Proceedings of the IEEE/CVF Conference on Computer Vision and Pattern Recognition (pp. 20331-20341).

**Strengths:**

1. This work proposes utilizing an event-based optical flow estimator to leverage the inter-frame event data as a denser motion supervision signal, thereby assisting in the optimization of the deformation field.

2. This work presents an unsupervised fine-tuning framework based on LoRA (low-rank adaptation) that uses contrast maximization to adjust a pretrained optical flow estimator.

3. By leveraging components such as projection, motion segmentation, and camera pose optimization, this work establishes an association between the 2D estimated optical flow and the motion of 3D Gaussians, thereby enabling effective supervision.

**Weaknesses:**

1. The base assumptions in this paper somewhat diverge from real-world conditions. Although event data, due to its high temporal resolution, can provide detailed inter-frame motion clues for large or rapid movements, the large motions (fast movements) would, in practice, result in motion blur in RGB frames captured by regular cameras. However, the experiments in the paper assume that sparse, clear RGB frames are available—a point that warrants reconsideration. Using event data to deblur RGB frames could be a potential solution.

2. Using 2D optical flow estimates obtained from events as supervision signals introduces substantial noise into the optimization process for 3DGS and deformation fields, as the optical flow estimates are not entirely accurate (and the authors did not analyze the intermediate results of the optical flow estimation in their experiments).

3. I have concerns about the plausibility of the authors' proposed method for fine-tuning a pre-trained optical flow estimator using the event flow from a single scene. First, the paper does not provide details on the dataset used for fine-tuning, nor does it offer any quantitative evaluation of the fine-tuned estimator. The qualitative comparison in the two presented frames lacks Ground Truth, rendering it unconvincing. Moreover, the Contrast Maximization (CM) framework [3] used by the authors is only a model-based event flow estimation method, and its performance is often unstable and not accurate enough in various situations. Thus, the claim that fine-tuning EvFlowNet [1] using CM can improve performance is questionable.

4. Further discussion reveals that LoRA [2] is typically used to fine-tune large models trained on massive datasets to adapt to specific domains or tasks. However, the rationale for applying LoRA [2] to fine-tune a small model focused on a specific task like optical flow estimation using event flow from a single scene is questionable. Additionally, the pre-trained EvFlowNet [1] model mentioned in the paper has not been trained on a broad dataset; it is generally trained in very limited scenarios, such as those encountered in autonomous driving or drone perspectives. Therefore, the performance of the fine-tuned optical flow estimator and its ability to generalize to different motion patterns are concerning.

5. The authors' method of directly applying Structure from Motion (SfM) to multi-view inconsistent image sequences for obtaining point cloud initialization may not be well-defined and has certain limitations. While it is true that 3DGS does not require a completely accurate initialization, it is important to note that a high proportion of dynamic elements in the scene or low texture in static areas can hinder SfM from providing a reasonable initialization point cloud. Consequently, this method may fail in some scenarios.

6. The selection of comparison pipelines in the authors' experimental section is not reasonable, thus lacking persuasiveness. Among the four baselines compared, only one is an event-based radiance field reconstruction method, while the others are based on RGB data. The authors should compare their method with more radiance field approaches that utilize both event and RGB modalities, such as EvDNeRF [4].

7. The experimental setup for comparisons with RGB-based methods may be unfair, as the paper does not detail the number of RGB frames used by different pipelines during evaluation. Since RGB-based methods do not benefit from additional event data supervision, relying solely on sparsely sampled RGB frames makes it difficult to achieve good performance. This may also explain why the performance of the RGB-based methods in this paper is generally lower than that reported in the original papers. Therefore, using the same number of sparse RGB sequences for all pipelines while allowing the authors' pipeline to utilize additional event information constitutes an unfair comparison. The authors should establish a more equitable experimental setup that enables the RGB-based pipelines to perform at their normal levels. This situation reflects the authors' underlying assumption that ordinary cameras can still capture clear RGB images during large or rapid movements, which limits the practical value of their pipeline. Conversely, if ordinary cameras could obtain dense and clear RGB sequences, the necessity of incorporating event data would warrant reconsideration. Event cameras typically demonstrate their advantages over traditional cameras under high-speed motion and low-light conditions.

[1] Zhu, A. Z., Yuan, L., Chaney, K., & Daniilidis, K. (2018). EV-FlowNet: Self-supervised optical flow estimation for event-based cameras. arXiv preprint arXiv:1802.06898.

[2] Hu, E. J., Shen, Y., Wallis, P., Allen-Zhu, Z., Li, Y., Wang, S., ... & Chen, W. (2021). Lora: Low-rank adaptation of large language models. arXiv preprint arXiv:2106.09685.

[3] Gallego, G., Rebecq, H., & Scaramuzza, D. (2018). A unifying contrast maximization framework for event cameras, with applications to motion, depth, and optical flow estimation. In Proceedings of the IEEE conference on computer vision and pattern recognition (pp. 3867-3876).

[4] Bhattacharya, A., Madaan, R., Cladera, F., Vemprala, S., Bonatti, R., Daniilidis, K., ... & Gupta, J. K. (2024). Evdnerf: Reconstructing event data with dynamic neural radiance fields. In Proceedings of the IEEE/CVF Winter Conference on Applications of Computer Vision (pp. 5846-5855).

**Questions:**

1. Are all the RGB image sequences used in the experiments clear images, free of motion blur?

2. Does the optical flow estimation pipeline need to be fine-tuned before reconstructing different scenes? If so, is the fine-tuned pipeline consistently better in performance than using EvFlowNet [1] or Contrast Maximization (CM) [2] for direct estimation?

3. When comparing the RGB-based methods and the methods utilizing both RGB and event data, was the same number of sparse RGB frames used for both? Is the event information considered as additional data beyond the same number of RGB frames?

4. Why compare only with one dynamic radiance field method that utilizes both RGB and event data, and not with more? Are all the evaluation results obtained from novel viewpoints and at novel times?

5. Could you provide qualitative and quantitative performance comparisons of the LoRA [3] fine-tuned optical flow estimator across different scenes with the original EvFlowNet [1] model and the Contrast Maximization (CM) method [2] ?

[1] Zhu, A. Z., Yuan, L., Chaney, K., & Daniilidis, K. (2018). EV-FlowNet: Self-supervised optical flow estimation for event-based cameras. arXiv preprint arXiv:1802.06898.

[2] Gallego, G., Rebecq, H., & Scaramuzza, D. (2018). A unifying contrast maximization framework for event cameras, with applications to motion, depth, and optical flow estimation. In Proceedings of the IEEE conference on computer vision and pattern recognition (pp. 3867-3876).

[3] Hu, E. J., Shen, Y., Wallis, P., Allen-Zhu, Z., Li, Y., Wang, S., ... & Chen, W. (2021). Lora: Low-rank adaptation of large language models. arXiv preprint arXiv:2106.09685.

---

### Note · Authors · 2024-11-28

**Comment:**

Thanks for chair and reviewer's time and attention, we are going to polish our method in the future step.

**Withdrawal Confirmation:**

I have read and agree with the venue's withdrawal policy on behalf of myself and my co-authors.